# Effectiveness of Displaying Traffic Light Food Labels on the Front of Food Packages in Japanese University Students: A Randomized Controlled Trial

**DOI:** 10.3390/ijerph20031806

**Published:** 2023-01-18

**Authors:** Nobuyuki Wakui, Raini Matsuoka, Chikako Togawa, Kotoha Ichikawa, Hinako Kagi, Mai Watanabe, Nobutomo Ikarashi, Miho Yamamura, Shunsuke Shirozu, Yoshiaki Machida

**Affiliations:** 1Division of Applied Pharmaceutical Education and Research, Faculty of Pharmaceutical Sciences, Hoshi University, 2-4-41 Ebara, Shinagawa-ku, Tokyo 142-8501, Japan; 2Department of Biomolecular Pharmacology, Hoshi University, 2-4-41 Ebara, Shinagawa-ku, Tokyo 142-8501, Japan

**Keywords:** food labels, food packaging, healthy eating, nutrition, traffic light food labels

## Abstract

Nutrition labeling on the front of food packages has been implemented worldwide to help improve public health awareness. In this randomized double-blind controlled trial, we used a Google Forms questionnaire to evaluate the effectiveness of nutrition labeling on food packages in university students. The questionnaire, ultimately completed by 247 students, included 15 dietary images from which they were asked to choose what they wanted to eat for breakfast, lunch, and dinner the following day. For the interventional (traffic light food [TLF]) group only, TLF labels were displayed on dietary images. This group had a significantly higher proportion of people conscious of healthy eating during all meals than the control group, and the effect of TLF labeling on choosing meals was the highest for lunch. In addition to the indicated nutritional components, the TLF group had a significantly higher proportion of people who were conscious of the ones of protein and dietary fiber that were not indicated on the label. The use of TLF labels resulted in an increase in the proportion of people choosing a healthy diet as well as being conscious of their nutritional components. Therefore, the use of TLF labels may help promote healthy dietary choices in Japan.

## 1. Introduction

The increasing prevalence of lifestyle-related diseases is a global concern. In particular, the incidence of obesity has been reported to be remarkably increased [1,2,3,4], and by 2025, one in five individuals of the world’s population will become obese if no effective action is taken [1,3,5]. Notably, obesity is a serious health problem as it causes many chronic diseases, such as type 2 diabetes and heart disease [2,6,7]. Thus, this condition leads to an increase in the number of critically ill patients in the community [8], significantly reduces life expectancy [9], and is even considered a public health threat [6]. For this reason, obesity prevention policies are being implemented worldwide as one of the important public health efforts [10,11,12].

As a specific policy, the method of labeling nutritional information on the front of food packages (front of pack [FOP]) is implemented mainly in European countries [13,14]. Through this method, lay people can select healthy foods spontaneously when purchasing foods. Traffic light food (TLF) labels, which were first developed in England [13,15], make healthy food choices easier, even for people with little knowledge of nutrition. Based on that merit, the label has been adopted as food labeling in many countries [16,17]. In addition, these labels have helped improve health awareness and may further help prevent lifestyle-related diseases [18,19,20,21].

Through the use of TLF labels, consumers can visually select healthy foods when shopping for groceries [18]. Notably, the labels of three colors (red, yellow, and green) on the signaling device help consumers to distinguish between healthy foods. In addition to the caloric labeling in foods, the components of total lipids, saturated fatty acids, sugars, and salts are indicated by one of the three colors [22,23]. These food components pose the greatest health risks when consumed in excess. Red is for large amounts, green is for small amounts, and yellow is for intermediate amounts. Energy (calories) is also displayed, but this is not color coded. Because it is easy to notice visually and can be judged at a glance, TLF labels may increase consumers’ health awareness when shopping for groceries, and may help to prevent lifestyle-related diseases, such as obesity.

Regarding the current situation in Japan, it has been reported that the incidence of obesity increases with age from the 20s [24]. The new food labeling system has been fully enforced from 1 April 2020 against the backdrop of the increasing health awareness of people, growing importance of nutrition labeling, and expansion of mandatory nutrition labeling overseas. As a result, nutrition labeling on foods has become mandatory [25], which requires the labeling of the energy, protein, fat, carbohydrates, and salt equivalents. Moreover, the labeling of saturated fat and dietary fiber is also recommended [26]. However, in Japan, nutrient content labels are described in foods, but these labels are difficult to understand because the descriptions are provided on the back of food packages. Unlike the TLF label, which provides visual nutritional information, the FOP labeling (FOPL) is also unfamiliar [27]. Furthermore, because the appropriate nutrient content needs to be judged by the individual, the description has not been used as effectively by the general public in daily life as in other countries [28]. A meta-analysis study based on overseas reports is conducted on the usefulness of FOP labeling by Japanese researchers [29], but an intervention study evaluating the effectiveness of nutrition labeling for Japanese people has not yet been conducted.

Therefore, we have conducted a randomized controlled trial for diet selection using Japanese university students to verify the effectiveness of the TLF nutrition label displayed on the front of food packaging. The specific research objectives were to determine (1) whether the use of TLF labels on the FOP increases people’s choices of healthy meals, (2) whether their effects are consistent across three meal timings in one day, and (3) whether they can increase people’s awareness about food components, such as calories, sugar, and saturated fats.

To the best of our knowledge, this is the first intervention study to examine the utility of FOPL in Japanese university students.

## 2. Materials and Methods

### 2.1. Study Design

This randomized double-blind parallel-group trial evaluated the impact of nutritional labeling on university students aged >20 years. All participants were asked to look at 15 dietary images and choose three meals from those images for breakfast, lunch, and dinner the following day. In addition, participants were randomly assigned to the TLF-nutrition-labeling (intervention) or no-TLF-nutrition-labeling (control) group. In the intervention group, the TLF labels were displayed on the FOP of dietary images, whereas in the control group, foods without the TLF labeling on the dietary images were selected in the same way as in the intervention group. We then compared the proportion of people who were conscious regarding healthy eating between the two groups based on the presence or absence of nutrition labels on the foods they chose.

All students agreed to participate in the study, and informed consent was obtained from all participants online before starting the survey. The study protocols were approved by the Ethics Committee of Hoshi University (Approval number: 2021-14).

### 2.2. Randomization and Masking

Participants were randomly assigned to one of the two groups (intervention or control) in a 1:1 ratio, and a diet selection survey was conducted in a double-blind manner so that participants and the interventionists, who were also the data analysts of the present study, could not discriminate between the groups of participants. Notably, randomization was performed using the permuted block method with a block size of 2. After being classified into two groups, participants were emailed a URL to a Google Forms survey in which they had to choose a meal based on whether they had a TLF label. The blinding was maintained until the end of the statistical analysis, with the participants responding independently to the URLs sent to them without discussing the contents of the survey.

### 2.3. Procedure

The present study did not involve contact between researchers and participants and was only conducted using Google Forms. The Food Choice Survey link was sent simultaneously to all participants at 14:00 on the day of the survey, and the participants responded to the survey between 14:00 and 15:00. The intervention group chose the meal they wanted to eat for breakfast, lunch, and dinner the following day from the images with a traffic light label used as a nutrition label in the UK. Alternatively, the control group chose their breakfast, lunch, and dinner the following day from the images with no TLF label. In addition, the intervention group chose the meal after reading a sentence explaining the importance of nutrition labeling while making the meal selection. The four ingredients on the nutrition label were total lipids, saturated fat, sugar, and salt, following the UK Food Standards Agency’s guidelines [13,30]. The images used for food selection were 15 common Japanese foods. In particular, the following 15 types of Japanese foods were used: (1) fried prawn bento, (2) rice ball bento, (3) fried oyster bento, (4) fried chicken bento, (5) grilled salmon bento, (6) salt-grilled mackerel bento, (7) ginger grilled pork bento, (8) sukiyaki bento, (9) dried laver bento, (10) croquette bento, (11) pork cutlet bento, (12) hamburger steak bento, (13) fried food bento, (14) grilled meat bento, and (15) various side dishes bento (Appendix A). The coloring of the nutrition labels in the dietary images was determined according to the university students’ preferred diet, which was identified in advance via a dietary selection survey administered to them. In particular, we conducted an “always better control” (ABC) analysis based on the results of a questionnaire survey in which the participants were asked to choose their favorite foods from the 15 Japanese foods used in the present study. The ABC analysis has been used in many studies and is based on Pareto’s rule of “significant minority and obvious majority” [31,32]. In this study, the meals were arranged in descending order of popularity, and the meals with a cumulative selection rate of 70% were colored red, 71–90% were yellow, and the 90–100% were blue. The less popular foods were labeled blue because the traffic light “go” signal in Japan is recognized as blue. In other words, we had set that the diet popular among students was an unhealthy diet, marked in red, and the nonpopular diet was a healthy diet, labeled in blue. Thus, the extent to which diets that are likely to be selected but are avoided in nutrition labeling can be determined by randomizing a comparison group.

In addition, in order to maintain a natural dietary choice situation, we did not make all the nutritional components in the same color. Instead of coloring all the nutritional components in the same color, we colored based on the results of the ABC analysis of the two components of the total lipids and salt [33], which are considered nutritional components that Japanese people are concerned about in their twenties. Other saturated fats and sugars were randomly assigned the color of blue or yellow. Furthermore, to describe the nutrient component values on the TLF labels, statistical software was used to generate random numbers within the color determination reference ranges on the labels, and the values obtained were described on the nutrient labels.

### 2.4. Setting the Nutrition Label

The color setting standards for nutrition labeling in the British Food Standards Agency guidelines used in this study are described below [30]. Total fat labeling was performed as follows: Meals containing >17.5 g total fat per 100 g were labeled red. Meals containing ≥3.1 g and ≤17.5 g total fat per 100 g were labeled yellow. Meals containing ≤3 g total fat per 100 g were labeled blue. Saturated fat labeling was performed as follows: Meals containing >5 g saturated fat per 100 g were labeled red. Meals containing ≥1.6 g and ≤5 g saturated fat per 100 g were labeled yellow. Meals containing ≤1.5 g saturated fat per 100 g were labeled blue. Sugar labeling was performed as follows: Meals containing >22.5 g sugar per 100 g were labeled red. Meals containing ≥5.1 g and ≤22.5 g sugar per 100 g were labeled yellow. Meals containing ≤5 g sugar per 100 g were labeled blue. Salt labeling was performed as follows: Meals containing >1.5 g salt per 100 g were labeled red. Meals containing ≥0.31 g and ≤1.5 g salt per 100 g were labeled yellow. Meals containing ≤0.3 g salt per 100 g were labeled blue.

### 2.5. Outcomes

The outcomes of this study involved determining (1) whether the proportion of people making healthy dietary choices increased in the TLF-label (labeled) group compared with the no-TLF-label (unlabeled) group; (2) whether the effect of nutrition labeling varied across the three choices for breakfast, lunch, and dinner; and (3) whether the proportion of participants who were aware of the dietary components (calories, carbohydrates, and lipids) differed between the labeled and unlabeled groups. Furthermore, after completion of this study, all participants answered the questionnaire on the validity of the TLF labels.

### 2.6. Statistical Analysis

The demographic characteristics of the participants were presented using descriptive statistics. We used mean and standard deviation to summarize the numerical data and frequency and ratio to summarize the categorical data. Fisher’s exact test was used to analyze (1) the effect of TLF labeling on the proportion of people who made healthy eating choices at each meal for breakfast, lunch, and dinner and (2) whether the proportion of people aware of the dietary components during food selection differed based on the presence or absence of a TLF label. In addition, we used the Cramer’s V score to assess (1) the magnitude of the effect of TLF labeling on each dietary component and (2) whether this magnitude varies with meal timing. All data were analyzed using the SAS software (version 9.4, SAS Institute, Inc., Cary, NC, USA). The number of missing values was 0 because we prepared the Google Forms in such a way that the survey would not end even if only one part is left unanswered. A *p*-value of < 0.05 was considered statistically significant.

## 3. Results

### 3.1. Participants

Overall, 274 students participated in the present study, and they were randomly assigned to the TLF-nutrition-labeling (labeled; n = 137) or no-TLF-nutrition-labeling (labeled; n = 137) group. Ultimately, 88.3% (n = 121) and 92.0% (n = 126) of participants in the labeled and unlabeled group responded completely to the survey, respectively, and a total of 90.1% (n = 258) participants responded completely (Figure 1).

The participants’ demographics and baseline characteristics were similar between the groups (Table 1). The overall mean age was 21.9 ± 1.1 years, most of the participants were women (69.6%), and the mean number of eating out per week was 1.8 ± 2.2. Regarding the comprehensibility of the nutrient content labeling on the back of foods used in Japan, most participants answered that the current nutrition labeling was difficult to understand (68.0%). Regarding the reference values for the dietary components, 68%, 31.2%, 23.1%, 13.8%, 9.3%, 85.8%, and 18.6% were the values for calories, protein, total lipids, sugars, saturated fatty acids, salt, and dietary fiber, respectively. Only a few people in either group understood the reference values, except for the calorie and salt reference values.

### 3.2. Impact on Selection of Foods with and without TLF Labels

The proportion of people who became conscious of healthy eating when choosing meals for breakfast, lunch, and dinner differed significantly between the labeled and unlabeled groups (Table 2). Regarding breakfast food choices, the proportion of people choosing yellow-labeled foods was 10% lower in the labeled group than in the unlabeled group, and the proportion of people choosing blue-labeled foods increased in the labeled group. Regarding meal choices for lunch, the proportions of people choosing red- and yellow-labeled foods were lower in the labeled group (15% and 14%, respectively) compared with those in the unlabeled group. Regarding meal choices for dinner, the proportions of people choosing red- and yellow-labeled foods were lower in the labeled group (16% and 10%, respectively) compared with those in the unlabeled group. Regarding the magnitude of the effect of nutrition labeling on meal timing, the Cramer’s V score for breakfast, lunch, and dinner was 0.19, 0.32, and 0.29, respectively, with labeling having the greatest effect at lunch.

### 3.3. Nutritional Awareness When Choosing Meals between Labeled and Unlabeled Group

Evaluation of people who were conscious of nutrition when choosing meals by the presence or absence of TLF labeling was conducted. The results revealed that 76.0% of participants in the labeled group and 36.5% of participants in the unlabeled group were conscious of some kinds of nutrition when choosing meals, and the difference between the two groups was significant (Table 3). In terms of the nutritional component, the proportion of people who were conscious of the nutritional component listed on the TLF label increased significantly. Moreover, significant nutritional awareness increases were also observed for protein and dietary fiber, which were not listed on the TLF label.

### 3.4. Questionnaire-Based Survey on TLF Labeling after the Completion of the Study

According to the results of the questionnaire, 98.4% and 96.0% of participants in the labeled and unlabeled groups, respectively, answered that colored nutrition labeling on the front of food packages improves health awareness (Table 4). Overall, 79.4% of participants responded that they wanted manufacturers to use colored nutrition labeling on the FOP. Additionally, 97.6% of participants agreed that colored nutrition labeling on the front of food packages is effective. In all surveys, the labeled group answered more positively about the use of TLF labels than the unlabeled group. In the question “How much did the colored nutrition labels on the front affect your dietary choices in this survey?”, which was answered only by the labeled group, 73.4% of the participants answered that these labels influenced their dietary choices on this occasion.

## 4. Discussion

In the present study, 274 university students were randomly classified into two groups using a food selection test with and without TLF labeling to investigate whether TLF labeling would lead to healthier dietary choices. The nutrition label was created with a traffic light label based on the British traffic light label color scheme. Due to this, the participants in the nutrition label group were able to make healthy diet choices quickly without possessing a thorough knowledge of nutrition labels. The present study did not have any major biases because the allocation of participants was randomized and comparability was ensured. The results showed that the use of TLF labels resulted in the avoidance of popular dietary choices among young people and that the proportion of people who made healthy dietary choices was significantly larger in the labeled group than in the unlabeled group. In addition, the proportion of people who were conscious of the nutritional components of foods when making dietary choices was higher in the labeled group than in the unlabeled group. Additionally, by placing the TLF label on the FOP, participants also became conscious of the nutritional components that were not included in the TLF label.

In addition, in the labeled group, the proportion of individuals who consumed a healthy diet at all meals was significantly higher in the labeled group than in the unlabeled group. In particular, the strongest effect of the label was observed at lunchtime, whereas the weakest effect occurred at breakfast. This could be attributed to the fact that the participants tend to choose light meals for breakfast in their everyday lives. When comparing lunch and dinner, the proportion of people who consumed red- and yellow-labeled meals for lunch was significantly lower (by more than 10%) in the labeled group than in the unlabeled group. Therefore, the timing of the choice of diet may also influence the awareness of healthy food choices. This study shows that the use of easy-to-understand nutrition labeling leads to healthier dietary choices among Japanese youth. This result seems to indicate the usefulness of the TL system applied in Western countries in Japan as well.

Regarding the question of whether the presence of a TLF label influenced the awareness of nutritional components when making meal choices, the proportion of people who were aware of nutritional issues was significantly higher in the labeled group (by approximately 40%) than in the unlabeled group. An interesting thing is the presence of the TLF labels resulted in a significant difference in the proportion of people who were conscious of the nutritional components of protein and dietary fiber, which were not indicated on the TLF label, between the two groups. Therefore, TLF labeling influenced not only the nutritional component listed on the label but also the awareness of the nutritional components not included on this label, thereby influencing the general awareness of trying to consume a healthy diet.

In the questionnaire-based survey on TLF labels conducted after the completion of this study, more than 90% of the participants answered that “TLF labels on the front of the packaging are effective”. Moreover, more than 75% of the participants answered that they “want manufacturers to put TLF labeling on the front of food packaging.” Approximately only 47% of the participants also answered that they use the current nutrient labeling in everyday life. Additionally, in the pre-test questionnaire, 68% of the participants answered that “the nutrient labeling on the back side of foods is difficult to understand.” The TLF label makes it easy to know the healthy amount of nutrients in a meal within a moment without the need for any calculations [13]; we believe that this convenience influenced the results. Therefore, using FOPL, such as TLF labeling, is beneficial for promoting healthy eating habits, even in Japan.

This study has some limitations. First, the lasting effects of the long-term use of TLF labels could not be assessed because only the meals that participants intended to eat the following day were recorded in this study. Second, because we only included college students, the effects of TLF labeling on people of other age groups remain unknown. Third, the results may be skewed due to gender influences as the proportion of female students was high (70%). Fourth, the participants did not actually eat their choice of food; therefore, it is not known whether these study results can be generalizable to eating behavior in the real world. These limitations must be considered when interpreting the results.

The TLF label feature reportedly signals participants’ brains to avoid red-labeled foods [34], similar to a red light signaling to stop a car. Moreover, the red signal labeling has been shown to potentially reduce unhealthy eating habits and activate the superior medial frontal gyrus and accessory motor area. Additionally, online surveys from other countries that do not have an FOPL have reported the usefulness of applying nutrition la-bels [35]. In the future, it would be useful to conduct an attribution analysis on the impact of TLF labels on Japanese consumers because there are reports that TLF use and nutrition-al knowledge are related to the socioeconomic and educational status of the participants [16]. Notably, the effect of FOP nutritional labeling was also reported in a foreign study involving children, in which the relative impact of FOP nutritional labeling on children’s choices of warning systems was greater than that of signaling systems. Therefore, it would be beneficial to conduct a comparative assessment of traffic light and other labels in future studies targeting Japanese people [36].

To the best of our knowledge, this is the first study to examine the benefits of FOPL in 274 Japanese university students, and this randomized double-blind controlled trial was conducted online and entirely independently, without the participants contacting each other. Given that participants answered the survey simultaneously, this study also avoided the effects of different response times. Interestingly, the presence of a nutrition label increased people’s awareness of the nutrients listed on the label as well as of the nutrients not listed on the label. In addition, a previous study reported that FOPL can promote healthier food purchases [14], and it is highly likely that it will be implemented in Japan and recognized for its effectiveness.

In general, consumers need to think relatively little when choosing and buying groceries as they generally do not have much time to do so; therefore, rapid processing and a simple FOPL has been shown to be effective [37]. When we examined the participants’ knowledge of the reference values for nutritional components, we found that few participants knew the reference values for nutrients other than calories, making it difficult to choose a healthy diet based solely on nutritional component labeling. This finding applies to many Japanese people because Japanese education does not teach the reference values for each dietary ingredient. In addition, upon the completion of this study, many participants expressed that the TLF label was useful. Overall, our study results suggest that the use of nutritional labels, such as TLF labels, can promote healthy dietary choices and help prevent lifestyle-related diseases in Japan.

## 5. Conclusions

We found that the use of TLF labels led to an increase in the proportion of people choosing a healthy diet. Moreover, the proportion of people who became conscious of nutritional components increased. Additionally, many participants revealed that the TLF label was useful, and they want manufacturers to put color nutrition labeling on the front of food packaging. Therefore, the use of nutrition labels, such as TLF labels, may help promote healthy dietary choices and prevent lifestyle-related diseases in Japan, similar to Europe and other countries.

## Figures and Tables

**Figure 1 ijerph-20-01806-f001:**
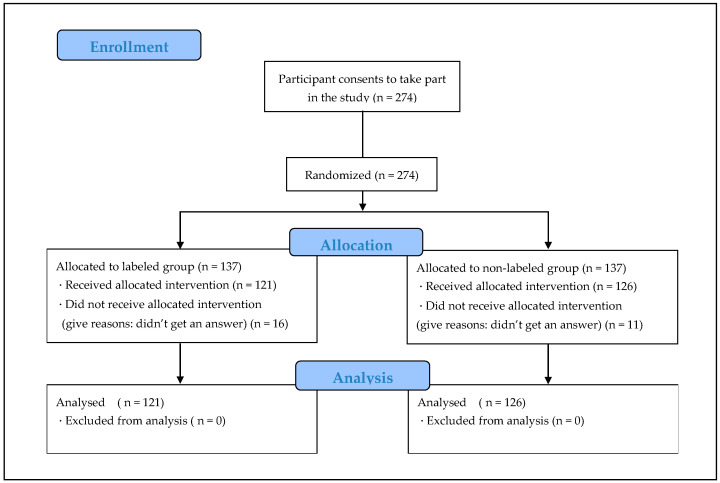
Flowchart of participant allocation.

**Table 1 ijerph-20-01806-t001:** Demographic and baseline characteristics of the study participants in the intervention and control groups.

	Labeled Group	Unlabeled Group
	(n = 121)	(n = 126)
Age	21.9 ± 0.8	22.0 ± 1.3
Sex (Female)	85 (70.3%)	87 (69.1%)
Frequency of eating out (per week)	1.6 ± 2.3	2.0 ± 2.0
Do you think the nutrient labeling on the back of foods is difficult to understand?	83 (68.6%)	85 (67.5%)
Do you know the reference intake of calories per day?	81 (66.9%)	87 (69.1%)
Do you know the reference intake of protein per day?	42 (34.7%)	35 (27.8%)
Do you know the reference intake of total lipids per day?	27 (22.3%)	30 (23.8%)
Do you know the reference intake of sugar per day?	20 (16.5%)	14 (11.1%)
Do you know the reference intake of saturated fatty acids per day?	15 (12.4%)	8 (6.4%)
Do you know the reference intake of salt per day?	102 (84.3%)	110 (87.3%)
Do you know the reference intake of dietary fiber per day?	25 (20.7%)	21 (16.7%)

Values for questions about the nutrient intake per day denote the number (percentage) of those who responded “Yes”.

**Table 2 ijerph-20-01806-t002:** Status of meal choices at breakfast, lunch, and dinner with and without TLF labels.

		Eat Nothing	Blue Label	Yellow Label	Red Label	Cramer’s V	*p*-Value
Breakfast						0.19	0.028
	Labeled group	13 (10.7%)	16 (13.2%)	91 (75.2%)	1 (0.8%)		
	Unlabeled group	10 (7.9%)	5 (4.0%)	108 (85.7%)	3 (2.4%)		
Lunch						0.32	<0.001
	Labeled group	2 (1.7%)	54 (44.6%)	44 (36.7%)	21 (17.4%)		
	Unlabeled group	0 (0%)	22 (17.5%)	63 (50.0%)	41 (32.5%)		
Dinner						0.29	<0.001
	Labeled group	0 (0%)	62 (51.2%)	31 (25.6%)	28 (23.1%)		
	Unlabeled group	1 (0.8%)	30 (23.8%)	45 (35.7%)	50 (39.7%)		

“Eat nothing” indicates those who answered that they did not eat at the specified meal times the next day. Labeled group, n = 121; unlabeled group, n = 126 at all meal times.

**Table 3 ijerph-20-01806-t003:** Nutritional awareness when choosing meals.

	Labeled Group	Unlabeled Group	Cramer’s V	*p*-Value
	(n = 121)	(n = 126)		
Calories	72 (59.5%)	40 (31.8%)	0.28	<0.001
Proteins	54 (44.6%)	29 (23.0%)	0.23	<0.001
Total lipids	76 (62.8%)	33 (26.2%)	0.37	<0.001
Sugar	48 (39.7%)	26 (20.6%)	0.21	0.0014
Saturated fatty acids	56 (46.3%)	26 (20.6%)	0.27	<0.001
Salt	72 (59.5%)	29 (23.0%)	0.37	<0.001
Dietary fiber	40 (33.1%)	18 (14.3%)	0.22	<0.001
Whole nutrition	92 (76.0%)	46 (36.5%)	0.40	<0.001

“Whole nutrition” refers to the question about whether the participants were conscious of some kinds of nutrition when choosing meals.

**Table 4 ijerph-20-01806-t004:** Post-experiment questionnaire about TLF labeling.

		Labeled Group	Unlabeled Group
		(n = 121)	(n = 126)
Q1	Do you want manufacturers to put color nutrition labeling on the front of food packaging?	101 (83.5%)	95 (75.4%)
Q2	Do you think that colored nutrition labels on the front are effective?	119 (98.4%)	122 (96.8%)
Q3	Do you usually use the nutritional information label on the back of the package?	57 (47.1%)	58 (46.0%)
Q4	Do you think that colored nutrition labels on the front improve health awareness?	119 (98.4%)	118 (93.7%)
Q5	Did the colored nutrition labels affect your dietary choices in this survey?	89 (73.4%)	-

Question 5 was answered only by the participants in the labeled group.

## Data Availability

The data are not publicly available as all participants have not consented to the public disclosure of the data online. However, the data presented in this study are available upon request from the corresponding author.

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
