# Peer review of "Effectiveness of Displaying Traffic Light Food Labels on the Front of Food Packages in Japanese University Students: A Randomized Controlled Trial"

_ijerph, 2023, doi:10.3390/ijerph20031806_

Round 1

Reviewer 1 Report

This manuscript described the effectiveness of front-of-package (FOP) food labels for making healthy dietary choices in Japanese university students. This research is novel because it is the first to examine FOP effectiveness in this population.

Introduction

·       Line 34: remove F from prevention

·       Line 40: First “particularly” can be deleted

·       Line 51: Temper this claim a bit by stating “may help to prevent”

Method

·       Can you provide example food and traffic light label images from the two conditions?

·       Line 99: What are some example Japanese foods that were used?

·       Lines 108-109, 115: Were less popular foods labeled blue or green? After looking this up, I have learned that traffic light “go” signals in Japan are blue! This might be helpful to describe for readers outside of Japan who are accustomed to green “go” signals.

Results

·       Figure 1 – “give reasons” for allocation seems like a placeholder that should be filled in

·       Lines 178-189: It’s unclear to me how awareness of nutritional components was measured. I know that participants chose foods for breakfast, lunch, and dinner. Were those choices also supposed to represent awareness? If so, I’m not sure if this represents awareness, and a better term might be something like choice. We can make choices without awareness of why we are making them (e.g., heuristics).

Discussion

·       Another limitation is that they did not actually eat their food choices, so it unknown if this would generalize to real world eating behavior.

Author Response

Dear Reviewer

Thank you for the thoughtful and constructive feedback you provided regarding our manuscript.

We agree with you that suggestion to change paper, and we have amended this by adding the content.

Please see the attached file for details. Thank you.

Sincerely,

Nobuyuki Wakui

Reviewer 2 Report

This paper examines TL's influences on food choice in Japanese college students. The paper's design is simple and  easy to understand. The choice might be influenced by a lot of demographic variables, however, the knowledge about TL information was not considered. By the way, the Japnese backgound about nutrition label use was not reported, so that the paper seems to be limited in deep our knowledge about the use of TL systems in a new environment. The paper also needs a revision in grammar. 

In section 2.3, I would really like to see an example of the picture about how you show your two groups of participants to choose their meals for the next day. Because the data is limited so that we did not see detailed pictures in this paper.

Maybe more relevant studies about TL should be cited in this paper, for instance in other developing countries where TL was not used but some pilot studies have investigated TL's influence on consumers' choices.

Author Response

(The authors gave the same response as above.)

Round 2

Reviewer 2 Report

In the last review, I suggest the author cite more work about the use and development of nutrition labels (e.g., nutrition panels ) in Japan, but the author seems to neglect my suggestions. In addition, the procedure is not clear to me. I do not know how different colors were used to indicate the health of foods. Furthermore, the relationship between the manipulation of the current study and the TL systems in western countries. 

Author Response

Dear Reviewer 2

I must sincerely apologize. We were misreading your last review points. As you pointed out, we have revised the manuscript. I also checked the entire manuscript. Thank you for pointing this out.

Sincerely, Nobuyuki Wakui
